# Potential Effects of Hyperglycemia on SARS-CoV-2 Entry Mechanisms in Pancreatic Beta Cells

**DOI:** 10.3390/v16081243

**Published:** 2024-08-02

**Authors:** Tara M. Michaels, M. Faadiel Essop, Danzil E. Joseph

**Affiliations:** 1Centre for Cardio-Metabolic Research in Africa, Department of Physiological Sciences, Faculty of Science, Stellenbosch University, Stellenbosch 7600, South Africa; tmichaels@sun.ac.za; 2Centre for Cardio-Metabolic Research in Africa, Division of Medical Physiology, Faculty of Medicine and Health Sciences, Stellenbosch University, Cape Town 7505, South Africa; mfessop@sun.ac.za

**Keywords:** SARS-CoV-2, COVID-19, hyperglycemia, new-onset diabetes, pancreatic β-cells, membrane fusion, endosomal entry

## Abstract

The COVID-19 pandemic has revealed a bidirectional relationship between SARS-CoV-2 infection and diabetes mellitus. Existing evidence strongly suggests hyperglycemia as an independent risk factor for severe COVID-19, resulting in increased morbidity and mortality. Conversely, recent studies have reported new-onset diabetes following SARS-CoV-2 infection, hinting at a potential direct viral attack on pancreatic beta cells. In this review, we explore how hyperglycemia, a hallmark of diabetes, might influence SARS-CoV-2 entry and accessory proteins in pancreatic β-cells. We examine how the virus may enter and manipulate such cells, focusing on the role of the spike protein and its interaction with host receptors. Additionally, we analyze potential effects on endosomal processing and accessory proteins involved in viral infection. Our analysis suggests a complex interplay between hyperglycemia and SARS-CoV-2 in pancreatic β-cells. Understanding these mechanisms may help unlock urgent therapeutic strategies to mitigate the detrimental effects of COVID-19 in diabetic patients and unveil if the virus itself can trigger diabetes onset.

## 1. Introduction

The coronavirus disease 2019 (COVID-19) pandemic is driven by the novel coronavirus, severe acute respiratory syndrome coronavirus-2 (SARS-CoV-2). Since January 2020, the World Health Organization has recorded approximately 775.6 million cases, with 7 million deaths due to COVID-19 [1]. These statistics illustrate that despite global efforts, COVID-19 remains a pressing health concern. Moreover, the constant emergence of novel SARS-CoV-2 variants hinders progress to eliminate threats posed by this virus.

Individuals infected with SARS-CoV-2 exhibit a wide range of symptoms, from asymptomatic to mild to severe illness [2]. Moreover, COVID-19 patients, either of advanced ages or with comorbidities such as diabetes mellitus (DM), were found to be predisposed to severe disease onset, often leading to ICU admission and/or mortality [3,4,5,6].

Coincident DM may be of particular concern in COVID-19 patients. For example, one study showed an associated increased risk of severity and mortality with DM [7], while others reported higher ICU admissions and higher mortality rates among diabetic COVID-19 patients compared to their non-diabetic counterparts [5]. Several other studies further reported significantly increased in-hospital deaths and severe disease outcomes associated with diabetes in COVID-19 patients [3,8,9,10].

Hyperglycemia, which is a defining characteristic of DM, has since been determined as an independent risk factor for COVID-19 severity and/or mortality [11,12]. Furthermore, several studies found that COVID-19 is associated with hyperglycemia in patients with and without known diabetes [13,14]. Patients who maintained euglycemia also showed improved outcomes of COVID-19 compared to those with hyperglycemia [15]. Another study further confirmed that a relatively high level of blood glucose upon hospital admission was an independent risk factor for the mortality of COVID-19 patients with diabetes [16]. Furthermore, improved glycemic control correlated with better outcomes in patients with COVID-19 and pre-existing type 2 DM (T2DM) [17]. These findings highlight the importance of monitoring and treating hyperglycemia, irrespective of diabetic history, to lessen the severity of COVID-19 progression.

In addition to the impact of hyperglycemia on COVID-19 severity, the emergence of new-onset diabetes (NOD) is of concern. NOD, which is also referred to as new-onset hyperglycemia, was reported in COVID-19 patients without any history of diabetes [10,18]. Several researchers have investigated cases of NOD, and Table 1 summarizes the key results from a selection of recent studies. It is important to note that this table is not exhaustive, and further research is warranted on this topic.

One explanation for the emergence of NOD cases may be the possibility that SARS-CoV-2 directly infects pancreatic β-cells which may result in dysregulated pancreas functioning and altered glucose metabolism in healthy individuals [25,26]. Recent evidence also showed that COVID-19 may cause direct damage to the pancreas, in addition to insulin resistance, which could exacerbate hyperglycemia in diabetic patients or result in NOD in previously non-diabetic individuals [27,28]. More specifically, SARS-CoV-2 could infect insulin-producing pancreatic β-cells and elicit its impairment [29,30].

The loss and/or dysfunction of β-cells is an underlying feature of DM, and the possibility that SARS-CoV-2 may induce similar effects alerts to an emerging health crisis. Pancreatic β-cells reportedly express most of the known SARS-CoV-2 entry receptors, such as angiotensin-converting enzyme-2 (ACE2) and Neuropilin-1 (NRP1) [29,31,32], hence rendering them vulnerable to SARS-CoV-2 infection [30].

These findings emphasize the bidirectional nature of diabetes and COVID-19. Not only does diabetes increase the risk of developing severe COVID-19, but SARS-CoV-2 is also capable of inducing diabetogenic effects. However, much remains to be elucidated about the exact alterations occurring in this bidirectional relationship. This review investigates the potential interplay between diabetes, more specifically hyperglycemia, and putative SARS-CoV-2 entry mechanisms into pancreatic β-cells.

## 2. The Bidirectional Relationship between Diabetes and COVID-19

Several studies demonstrated that elevated glucose levels upon hospital admission were an independent risk factor for the progression of severe COVID-19 cases and mortality [11,33]. For example, some researchers found that 40% of their subjects displayed uncontrolled hyperglycemia upon admission and that in-patient mortality was greater for patients with diabetes [34]. They also noted that mortality was higher for patients without pre-existing diabetes who developed hyperglycemia in the hospital. These findings were further strengthened by Lazarus et al. [35], who showed a relationship between admission fasting glucose levels and COVID-19 severity. Overall, this highlights the important role of hyperglycemia in SARS-CoV-2 infection and, subsequently, the severity of COVID-19 onset.

COVID-19 can also affect glucose homeostasis in individuals without a history of diabetes [36]. The accompanying increase in cytokine production and dysregulated metabolic responses observed during severe COVID-19 may contribute to the onset of insulin resistance and subsequent hyperglycemia [37]. Recent findings also suggest a direct effect of SARS-CoV-2 on glucose metabolism as NOD, diabetic ketoacidosis, and unusually high insulin requirements to achieve glycemic control were reported for COVID-19 patients [14,17,22,26,38,39,40,41].

A plausible explanation for metabolic dysregulation during SARS-CoV-2 infections may be due to the direct damage of pancreatic β-cells to thereby impair insulin secretion [42]. Pancreatic endocrine cells, specifically β-cells, are responsible for insulin synthesis, storage, and secretion in response to changes in blood glucose levels [43]. To be able to respond to high glucose levels, glucose transporter 2 (GLUT2) is utilized for glucose uptake into β-cells. Glucose is then metabolized via glycolysis and, subsequently, oxidative phosphorylation to generate ATP. This increases the ATP/ADP ratio, leading to the closure of the ATP-sensitive K^+^ channels and the depolarization of the membrane potential. Ca^2+^ channels open because of this depolarization, allowing for extracellular Ca^2+^ ions to enter the β-cells and induce the exocytosis of secretory vesicles to secrete insulin [44].

An immune response via the release of cytokines and chemokines mediated by SARS-CoV-2 may induce damage in β-cells and attenuate their functionality through a loss of the ability to sense glycemic levels and insulin release [14,26,45]. Multiple studies reported cases of acute pancreatitis or pancreatic injury in patients who developed post-SARS-CoV-2 infections [46,47,48,49,50,51,52,53]. This emphasizes the ability of SARS-CoV-2 to directly affect the endocrine system, which of course includes the pancreas [54,55,56].

SARS-CoV-2 enters host cells via its primary receptor, ACE2 [57,58,59], with its expression increased upon inflammatory stress. This suggests an enhancement of β-cell sensitivity to SARS-CoV-2 infection during inflammatory conditions [60] such as DM. In agreement, there was increased ACE2 expression together with a pro-inflammatory profile in patients with diabetes and COVID-19 [61]. An upregulation of ACE2 receptors was also postulated to favor viral entry into host cells, which results in a relatively higher viral load and poor prognosis. Following viral infection, the loss of ACE2′s physiological function may enhance systemic adverse effects of the renin–angiotensin–aldosterone system (RAAS) [62].

Various studies revealed that ACE2 participates in the regulation of metabolism and, specifically, glucose homeostasis. For example, ACE2 deletion worsened glucose tolerance in their non-obese diabetic mice models, eventually resulting in NOD [63]. Moreover, infusion of angiotensin-2 (Ang2) in a C57bl/6J mouse model, which is cleaved by ACE2 to form angiotensin-1-7 (Ang1-7), results in decreased ACE2 pancreatic activity and expression accompanied by increased angiotensin type-1 receptor (AT1R) expression and oxidative stress [64]. Furthermore, ACE2 deletion increases oxidative stress and ACE and AT1R expression in pancreatic islets [63].

Overall, these findings confirm the role of ACE2 (SARS-CoV-2 receptor) in causing metabolic dysregulations, especially regarding glucose control and insulin secretion. The controversy surrounding ACE2 expression in β-cells leads to the hypothesis that other receptors, such as NRP1, Dipeptidyl peptidase-4 (DPP4), aminopeptidase N (APN), and Basigin, may be recruited in terms of SARS-CoV-2 entry mechanisms. Other proteases, such as members of the A disintegrin and metalloproteinase (ADAM) family, may also be recruited together with or in the absence of the primary SARS-CoV-2 protease, transmembrane protease serine-2 (TMPRSS2). Nevertheless, it is evident that there is a potential bidirectional relationship between COVID-19 and diabetes.

## 3. Unveiling the Viral Structure and Function of SARS-CoV-2

Coronaviruses (CoVs) are a family of enveloped, positive-strand RNA viruses that can cause respiratory and enteric disease in animals and humans [65]. These viruses are becoming an increasing health concern following three recent zoonotic outbreaks of highly pathogenic human CoVs (HCoVs), namely the severe acute respiratory syndrome coronavirus (SARS-CoV) from 2002 to 2003, the Middle East respiratory syndrome CoV (MERS-CoV) in 2012, and the current SARS-CoV-2 pandemic that began in 2020 [66,67,68,69].

The general structure of CoVs includes four structural proteins: the spike (S), envelope (E), matrix (M), and nucleocapsid (N). The E- and M-proteins play central roles in viral assembly, while the N-protein is required for the virus’s RNA synthesis [70]. The S-protein mediates receptor binding and the fusion of viral particles with target host cells. Figure 1 illustrates the general structure of a CoV, with a specific focus on the S-protein.

The S-protein is a type 1 fusion protein that functions as a trimer. Each monomer is divided into two subunits: S1, which mediates receptor binding, and S2, which contains the transmembrane domain and mediates fusion with the host cell membrane [71]. The receptor binding domain (RBD) is found within the S1 subunit and is responsible for binding to receptors of the CoVs [72]. The S-protein also undergoes post-translational cleavages by host cell proteases to expose this RBD to host cell receptors and allow viral entry.

Proteases involved in CoV S-protein activation can act at different stages of the viral life cycle. These include proprotein convertases (Furin) during viral packaging, cell surface proteases (transmembrane serine proteases) after receptor binding, and lysosomal proteases (Cathepsins B/L) after virion endocytosis [73,74]. The role of S-protein priming by host cell proteases is particularly important in SARS-CoV-2 viral entry and tropism. The S1/S2 junction is cleavable by either trypsin-like proteases (i.e., TMPRSS2) that are present on the host cell membrane and endosomal cathepsins that are activated by a drop in pH (i.e., Cathepsins B/L) [59,75].

The different types of S-protein priming may be crucial factors in explaining SARS-CoV-2 cell tropism and the features of COVID-19 symptoms [76]. The type of protease priming may also determine whether the membrane fusion process occurs directly at the plasma membrane or endosomal levels [77]. Several studies showed that both TMPRSS2 and Cathepsins B/L are important for SARS-CoV-2 entry [59,78]. Figure 2 summarizes the key events in each main entry pathway, namely the membrane fusion and the receptor-mediated endosomal entry pathways.

## 4. Membrane Fusion Entry Pathway

The SARS-CoV-2 infection of host cells is completed in four steps: (1) prebinding, (2) receptor binding, (3) proteolytic cleavage, and (4) membrane fusion. Before SARS-CoV-2 can enter cells, S-proteins exhibit a prebinding conformation with an exposed RBD to enable its attachment to host membrane receptors [80], such as ACE2, in the second step. Upon binding to the receptor, S-proteins are cleaved by proteases (TMPRSS2) present in the host cell membrane [81] to reveal the S2 subunit through the shedding of the S1 subunit [82] and activating membrane fusion. After the final step, when the cell membranes of the virus and host are fused, SARS-CoV-2 releases its viral material into the host cell cytosol and initiates other infectious processes [83,84].

To understand how SARS-CoV-2 gains entry into pancreatic β-cells, we previously discussed the mechanisms of viral membrane fusion. Now, we will explore how hyperglycemia can influence proteins involved in membrane fusion, potentially enhancing the susceptibility of β-cells to infection. We also highlight other potential receptors and proteases that may be part of the SARS-CoV-2 membrane fusion entry into β-cells.

### 4.1. Angiotensin-Converting Enzyme-2

The primary receptor for SARS-CoV-2 is ACE2, the plasma membrane-bound carboxypeptidase [57,59,85]. SARS-CoV-2 S1 subunits induce ACE2 downregulation upon binding. Cells combat this viral infectivity toward adjacent cells by decreasing the amount of ACE2 on their surfaces [86]. This results in an overall lowering of ACE2 activity, leading to marked elevations in Ang2 with a reduction in Ang1-7 that induces pancreatic dysfunction, insulin secretion inhibition, and resultant hyperglycemia [87].

While a homolog of the angiotensin-converting enzyme (ACE), ACE2 possesses recognized roles in the RAAS [88]. The RAAS is an endocrine system that regulates blood pressure, fluid, electrolyte balance, and volume homeostasis through active metabolites [89]. ACE2 is a potent negative RAAS regulator that is crucial for maintaining the system’s homeostasis [90,91]. The main function of ACE2 is to cleave Ang2, which has potent vasoconstriction, pro-inflammatory, and pro-fibrotic effects, into Ang1-7 to induce vasodilatory, antiproliferative, and apoptotic effects [88,90,92,93]. Ang2 binds to AT1R and angiotensin type-2 receptor (AT2R), and Ang1-7 binds to the Mas receptor [94] to exert its antagonistic effects.

Tissues such as the lungs, kidney, small and large intestine, and vasculature express ACE2 [95,96,97]. The expression of ACE2 in the pancreas, however, has been contradictory. ACE2 can be expressed in β-cells, acinar cells, ductal cells, and in the islet microvasculature and pericytes [32,52]. Variation in ACE2 localization within islets was demonstrated in deceased COVID-19 patients [98], indicating differences in severity and outcome reported as well as contradictions between studies. Several studies reported that ACE2 is expressed in both endocrine and exocrine pancreatic tissues, specifically in the islets of Langerhans [29,99,100,101,102,103]. In the human pancreas, ACE2 is highly expressed in β-cells compared to other islets, in addition to pericytes and endothelial cells surrounding the islets [60].

Contrary to these findings, some studies found that β-cells do not express ACE2 [52,104,105], leading to the hypothesis that ACE2-dependent entry of SARS-CoV-2 is questionable in β-cells. This also highlights the potential coordination of ACE2 with other surface receptors as entry factors for SARS-CoV-2 invasion into the pancreas.

The discrepancies between research work that identified ACE2 in β-cells and those that did not could be explained by differences in experimental approaches and contexts for such studies. For instance, Fignani et al. [60] reported ACE2 in some islet endocrine cells from seven non-diabetic donors and one T1DM donor. However, ACE2 was mainly observed in the cytoplasm with a small fraction on the plasma membrane. These same authors utilized three ACE2 antibodies that recognize different epitopes of the full-length ACE2 in immunofluorescence staining. Coate et al. [104] employed two of these antibodies and did not detect either form of ACE2 in β-cells. These two studies, therefore, reveal how differing protocols may be a source of variability in terms of protein expression. Later sections of this review will highlight controversies surrounding the expression of other proteins in β-cells and why this may be the case.

Regarding the role of ACE2 in the pancreas, local RAAS in the islets regulates glucose homeostasis through ACE2 and Ang1-7 [106]. ACE2 overexpression in the pancreas of T2DM mice improved glucose tolerance and increased insulin secretion and β-cell proliferation [107]. ACE2 deficiency can also impact islet development and β-cell function [108,109]. Moreover, Ang2 can impair insulin signaling and islet function [110,111,112,113]. The ACE2/Ang1-7/Mas axis within β-cells is involved in the proliferation, differentiation, functioning, protection, and insulin production [114,115,116,117]. These findings emphasize the crucial role of ACE2 in islet functioning.

Uncontrolled hyperglycemia observed in patients with DM could also regulate ACE2-S-protein binding [118,119], possibly through the glycosylation and glycation of ACE2. Glycosylation is a post-translational process whereby glycans are enzymatically attached to macromolecules [120,121,122], whereas glycation is the non-enzymatic attachment of glucose to macromolecules [123]. ACE2 is a metalloprotease with a relatively long half-life and can hence be easily glycated by enzymatic and non-enzymatic mechanisms in patients with chronic hyperglycemia. This would favor the formation of advanced glycation end-products (AGEs) [118,124] that were linked to an increased risk for COVID-19 [125].

In accordance, relatively high total and glycosylated ACE2 expression levels were detected in COVID-19 heart samples from diabetic patients compared to non-diabetic samples [118]. Interestingly, one in silico study reported that the glycation of residues in both the S-protein and ACE2 resulted in a loss of interaction between them [119]. Considering this, alternative viral entry proteins may be involved [126] as viruses can develop new effective ways to enter host cells when their main entry mechanism is impaired [127,128].

### 4.2. Dipeptidyl Peptidase-4

Another potential entry factor is DPP4, or Cluster of Differentiation (CD) 26, which functions as a receptor for SARS-CoV-2 as it is the primary receptor for MERS-CoV [129,130,131]. Some researchers predicted that the S-protein of SARS-CoV-2 directly interacts with DPP4 in host cells [132], while another demonstrated how the RBD of the S-protein directly binds to DPP4 [133]. In contrast, some studies report that DPP4 may not be used for cellular entry by SARS-CoV-2 [57,59].

A molecular docking study by Cameron et al. [134] showed that the SARS-CoV-2 S-protein RBD interaction with DPP4 may be significantly weaker compared to that of the MERS-CoV-DPP4 complex, attributed to the differences in amino acid sequences between the two viral S-protein RBD regions. However, a later docking study showed that mutations in the RBD of the delta variant may enhance the SARS-CoV-2-DPP4 interaction to a similar degree compared to that of MERS-CoV [135]. These authors concluded that the more virulent variants may be better suited for DPP4 binding.

There is increased DPP4 expression in airway samples from COVID-19 patients [136,137], while its inhibition was associated with lower COVID-19 mortality [138]. Considering that DPP4 is a highly proteolytic protein, it was postulated as a protease for the cleavage of SARS-CoV-2 S-proteins [132]. Although the role of DPP4 as a receptor and protease is not yet fully elucidated, these findings highlight the potential function of DPP4 in SARS-CoV-2 entry mechanisms.

It is also a proteolytic enzyme that is involved in the degradation and inactivation of incretins, a group of gut hormones that stimulate insulin secretion [139,140]. These include glucagon-like peptide-1 (GLP-1) and glucose-dependent insulinotropic polypeptide (GIP) that function to augment glucose-induced insulin release from pancreatic β-cells, suppress glucagon secretion, and slow gastric emptying [141,142]. Inhibition of DPP4 can improve glycemic control and exert the beneficial effects of GLP-1 and GIP [143]; for example, the former positively stimulates the proliferation of β-cells to regulate their cell mass [144].

According to the available research, the expression of DPP4 in islets varies, similar to that of ACE2. Some studies support its expression [98,145], while others dispute it [146,147]. Interestingly, Shah et al. [148] found that DPP4 was detected in conditioned medium of human islets, suggesting that DPP4 is released from islets and also present in them.

Fadzeyeva et al. [149] found that a mouse model systemically lacking DPP4 displayed improved islet health and glucoregulation compared to their wild-type counterparts. The direct inhibition of DPP4 in human islets improved β-cell function, survival, and insulin secretion [150,151,152]. Therefore, an increase in DPP4 could translate to a loss of β-cell mass and attenuated insulin secretion.

Emerging evidence suggests that there is a crosstalk between DPP4 and the RAAS. For example, Ang2-infused mice kidneys displayed increased DPP4 activity [153]. These authors also found that supraphysiological Ang2 concentrations stimulated DPP4 activity in cultured proximal tubule cells. Similar findings were reported by Lee et al. [154] in podocytes that may apply to other cell types, such as β-cells. Given that the RAAS in the islets regulates glucose homeostasis [106], it is plausible that an increase in DPP4 results in impaired insulin secretion through the increase in Ang2 [155].

Relatively high levels of DPP4 are observed in patients with T2DM compared to controls [156]. Following this, hyperglycemia could stimulate DPP4 mRNA expression and enzyme activity in human glomerular endothelial cells [157]. Mannucci et al. [158] presented data showing that the degree of hyperglycemia in both T1DM and T2DM directly correlates with circulating DPP4 activity. Additionally, they showed that alterations in DPP4 were correlated with variations in HbA_1_c. These findings suggest that chronic hyperglycemia could stimulate DPP4 activity. This could result in a reduction in active GLP-1 levels, possibly contributing to hyperglycemia.

Evidence suggests a positive correlation between DPP4 elevations in diabetic and SARS-CoV-2-induced effects in host cells and its potential role in facilitating viral entry. In addition, the increase in DPP4 results in enhanced degradation of GLP-1, which can stimulate β-cell proliferation. A lack of GLP-1 could decrease β-cell mass and attenuate insulin secretion, with hyperglycemia as the overall outcome. This interplay between dysregulated DPP4 and hyperglycemia may contribute to developing NOD in susceptible individuals.

### 4.3. Neuropilin-1

Neuropilin-1 (NRP1) is a cell surface receptor involved in angiogenesis, organ development, and immune function [159,160]. It possesses a large extracellular domain that binds ligands in several signaling pathways associated with cell migration, growth, and development [161].

NRP1 was recently identified as a potential receptor for SARS-CoV-2 entry [31,162]. Some researchers found that SARS-CoV-2 utilizes NRP1 as a receptor in β-cells in the absence of ACE2 [29], which would lead to β-cell apoptosis and a reduction in insulin secretion. These authors also reported upregulated NRP1 levels in β-cells from individuals with COVID-19. Islet NRP1 is highly expressed, mostly confined to β-cells and rarely in pancreatic α-cells [29,30]. In contrast, Ji et al. [163] found that NRP1 was weakly expressed in islets. While the presence of NRP1 in pancreatic islets cannot be entirely ruled out, the lower levels observed by Ji et al. [163] suggest a potentially limited role in this specific tissue. More research is warranted to definitively determine NRP1’s involvement in SARS-CoV-2 infection and its expression patterns across various human tissues.

SARS-CoV-2 S-protein cleavage, mediated by Furin, results in a C-terminal motif on the S1 subunit which binds to NRP1 [31,162,164]. NRP1 stabilizes this C-terminus to allow for more efficient S1/S2 cleavage and to enable the S2 subunit to mediate membrane fusion more rapidly [164]. Inhibiting the binding between this C-terminal motif and NRP1 results in reduced viral infection in various cell types, including human islets [31,162,164].

Of note, research indicates that NRP1 may be a co-receptor for ACE2. For example, NRP1-knockout (KO) ACE2-expressing HeLa cells displayed less viral entry than those expressing ACE2 and NRP1 [162]. Furthermore, human islets infected with SARS-CoV-2 ex vivo showed that ACE2- and NRP1-expressing cells had more SARS-CoV-2 N-protein transcripts than those expressing either ACE2 or NRP1 [30]. Furthermore, these authors found that ACE2-expressing NRP1-KO cells contained fewer RNA transcripts than the ACE2-KO NRP1-expressing cells, while the double-KO cells exhibited little to no infection. Together with Cantuti-Castelvetri et al. [31] who reported on the independent facilitation of SARS-CoV-2 S-protein pseudoviral entry by ACE2 but not TMPRSS2 or NRP1, it can be postulated that NRP1 is a co-receptor that potentiates ACE2-mediated SARS-CoV-2 infection [165].

The role that NRP1 plays in the context of diabetes is controversial. Minor alleles of NRP1 have been associated with T1DM in children [166], which suggests that it could influence the development of some T1DM cases. Moreover, an analysis of a (cryopreserved) human diabetic kidney RNA sequencing dataset revealed that NRP1 was significantly upregulated [167]. However, others demonstrated decreased NRP1 protein expression in cultured differentiated podocytes, podocytes from diabetic db/db mice, and diabetic patients diagnosed with diabetic nephropathy [168,169]. Further studies are therefore needed to help navigate such contradictory findings and to demonstrate whether the upregulation and participation of NRP-1 in COVID-19 could result in long-term complications.

### 4.4. Glucose-Regulated Protein-78

Glucose-regulated protein-78 (GRP78), also referred to as heat shock protein A5, is an essential endoplasmic reticulum (ER)-localized chaperone that regulates ER signaling molecules to ensure proper protein folding or direct unfolded proteins to cellular degradation systems [170,171,172].

Viral infection may be promoted by GRP78 by its functioning as an alternative receptor and/or stabilizing S-protein binding [173]. For example, research work predicted that GRP78 can dock at the RBD of the SARS-CoV-2 S-protein [171], which has the same GRP78 recognition site as other human and bat CoVs [174]. Han et al. [175] provide evidence that cell surface GRP78 is utilized by SARS-CoV-2 to enter cells that express low or no ACE2, such as monocytes and macrophages. Considering the variable expression of ACE2 in β-cells (discussed in Section 4.1), SARS-CoV-2 may use GRP78 to enter these cells in an ACE2-independent manner. GRP78 also binds to the MERS-CoV S-protein in vitro, and some researchers showed that GRP78 knockdown attenuated MERS-CoV infection [176]. Based on such evidence, it can be postulated that GRP78 is involved in the entry mechanisms of SARS-CoV-2.

SARS-CoV-2 infection induces ER stress in host cells [177,178], possibly due to the inability to accommodate the significant increase in protein production that occurs during viral replication [179]. CoV infections can upregulate GRP78 levels [176,180]. For example, research studies found increased GRP78 serum levels in COVID-19 patients compared to COVID-19-negative pneumonia patients and healthy individuals [177,181].

Other potential SARS-CoV-2 host entry factors can also modulate GRP78 levels under conditions of ER stress. For instance, Basigin mediates ER stress-induced GRP78 upregulation [182]. This indicates that Basigin upregulation following ER stress may exacerbate GRP78’s mediation of viral entry. GRP78 can also influence cellular ACE2 levels; for example, Carlos et al. [183] observed decreased cell surface ACE2 levels, but not total ACE2, due to GRP78 knockdown (independent of ER stress). This suggests a possible role for GRP78 in ACE2 trafficking to the cell surface.

GRP78 is expressed in all pancreatic cell types, particularly β-cells [147,184]. It has also been reported that GRP78 increased in diabetic mouse models [185,186] and islets from donors with T2DM compared to non-diabetic donors [185,187]. Others also noted significantly increased serum concentrations of GRP78 in T2DM patients, with positive correlations between GRP78 levels and HbA_1c_ and AGEs [188]. In contrast, some studies have found no significant increases in islet GRP78 levels in T2DM donors [189,190].

Chronic hyperglycemia disrupts ER homeostasis to cause ER stress [191]. This could increase the levels of GRP78 within β-cells and provide an additional entry for SARS-CoV-2. The same outcome of increased GRP78 levels could be achieved with SARS-CoV-2 inducing ER stress. As reviewed by Cao et al. [192] and Lee et al. [193], ER stress plays an important role in β-cell dysfunction and loss, resulting in augmented insulin responses to hyperglycemia as observed during diabetes. Therefore, the shared mechanisms between hyperglycemia, SARS-CoV-2, and GRP78 may offer valuable insights for developing therapeutic strategies to target ER stress to improve patient outcomes.

### 4.5. Transmembrane Protease Serine-2

TMPRSS2 is a type 2 transmembrane serine protease located on the surface of host cells and whose expression is influenced by androgen [194]. Besides its role in cancer, particularly prostate cancer [195,196], its function in other pathologies was relatively unknown until SARS-CoV emerged. TMPRSS2 activates the S-protein of SARS-CoV for membrane fusion of the viral and host membranes [197,198,199]. Research has established that this protease activates S-proteins of SARS-CoV-2 to allow entry via membrane fusion into host cells [59]. For example, SARS-CoV-2 S-protein-mediated cellular entry was attenuated through in vitro androgen deprivation or antagonism [200].

According to RNA and protein expression data retrieved from the Human Protein Atlas database [201], TMPRSS2 is mainly expressed in the lungs, salivary glands, thyroid, gastrointestinal tract, kidneys, liver, and pancreas. As with ACE2 and DPP4 expression, the expression of TMPRSS2 in endocrine β-cells is controversial. Here, some studies support the notion that TMPRSS2 is indeed expressed in β-cells [17,29,32,98,101], whereas others concluded that its expression is either extremely low or undetectable [104,105]. Of note, Piva et al. [202] observed that TMPRSS2 expression is higher in pancreatic β-cells of T2DM patients compared to controls.

Some researchers found relatively high expression levels of TMPRSS2 in diabetic COVID-19 heart sample autopsies compared to non-diabetic ones [118]. It has also been highlighted that SARS-CoV-2 infection-related damage to pancreatic tissues is mediated by TMPRSS2 [203], suggesting a fundamental role for the virus in disease outcomes with diabetes. This is a possible explanation for why DM is an important risk factor for hospitalization and death in COVID-19 patients. However, Zhang et al. [204] reported that elevated glucose levels did not affect TMPRSS2 levels.

While some studies suggest that relatively high glucose conditions may elevate TMPRSS2 expression, the results remain inconclusive. Despite its potential glucose dependence, TMPRSS2 is a crucial protease for SARS-CoV-2 entry by facilitating S-protein cleavage at the cell surface. However, it is not the sole role player, and other proteases, such as ADAM10 and ADAM17 [205], can also cleave the S-protein. This can be potentially competitive with TMPRSS2, as observed for SARS-CoV [206]. This redundancy in the cleavage process highlights the adaptability of SARS-CoV-2 and the potential for therapeutic strategies that target not just TMPRSS2, but a broader range of host cell proteases involved in viral entry.

### 4.6. A Disintegrin and Metalloproteinase Family

The ADAM family includes a group of multidomain transmembrane proteolytic enzymes that are involved in regulating many physiological processes such as cell adhesion and migration, intracellular signaling, the regulation of growth factors and cytokines, and proteolysis of the intracellular matrix. These processes maintain homeostasis within the body but also play an important role in pathophysiologic complications such as diabetes [207,208]. ADAM9, ADAM10, and ADAM17 are prominent proteases involved in various physiological processes such as development, regeneration, immunity, and the regulation of cell differentiation and proliferation through cleaving ligands and receptors [209,210,211,212,213,214].

ADAM9 was identified as a potential host factor for SARS-CoV-2 S-protein-mediated infection [215] as ADAM9 knockdown resulted in significantly diminished SARS-CoV-2 infection. The authors also highlighted how the use of ADAM9 in SARS-CoV-2 entry in low-expressing ACE2 cells primarily depends on the latter. Although ADAM9 is associated with SARS-CoV-2 entry, it does not cleave the S-protein. Some showed that ADAM9 directly interacts with the RBD of the SARS-CoV-2 S-protein [216] which shifts its role from a potential S-protein protease to a potential receptor.

Of note, ADAM9 was identified as a key driver of SARS-CoV-2 severity in a multi-omics analysis of a young, comorbidity-free COVID-19 patient cohort [217]. This analysis found that increased ADAM9 RNA expression levels were detected in the serum of critically ill COVID-19 patients, further emphasizing the role that ADAM9 plays in SARS-CoV-2 infection.

Jocher et al. [205] established the role of both ADAM10 and ADAM17 as host cell factors for SARS-CoV-2 entry, as it cleaves the S-protein in a TMPRSS2-independent manner. Yamamoto et al. [218] found that SARS-CoV-2 pseudoviral entry was significantly inhibited using an ADAM10 knockdown model. Furthermore, circulating levels of ADAM17 were increased in patients with COVID-19 [219]. Inhibition of ADAM10 significantly blunts SARS-CoV-2 infection, whereas ADAM17 did not [218]. This demonstrates the potential role of ADAM10 and ADAM17 in SARS-CoV-2 viral entry, where SARS-CoV-2 entry is more dependent on ADAM10 than ADAM17.

Both ADAM10 and ADAM17 can shed ACE2 [206,220,221,222,223] and promote SARS-CoV-2 pathogenesis [224]. In contrast, ADAM9 does not possess the ability to cleave ACE2 [220]. The proteolysis or shedding of ACE2 releases a soluble, enzymatically active form corresponding to the ACE2 ectodomain [222,225,226]. While the function of soluble ACE2 (sACE2) is still obscure, the current literature shows that the shedding mechanism is under strict molecular control [227]. Patel et al. [228] identified that Ang2 activates ADAM17, which increases the release of sACE2. Other research work further confirmed ADAM17’s role in regulating ACE2, where its overexpression increased ACE2 shedding and subsequently decreased cellularly bound ACE2 in mouse pancreatic islets [229].

Multiple studies reported ADAM9’s expression in Langerhans’s islets [230,231], while others found that ADAM17 is expressed in islets, including β-cells [104,147,184]. ADAM10 is present in the pancreas [232] and is localized in the cell membranes of endocrine and exocrine cells [231].

Regarding the effects of hyperglycemia on ADAM9, Puddu et al. [233] found increased expression under high glucose conditions in a retinal pigment epithelial cell model. Similarly, Moin et al. [234] found that ADAM9 levels were elevated in T2DM, while Suresh Babu et al. [235] reported increased ADAM9 expression in macrophages exposed to high glucose and in human diabetic hearts.

C-X-C motif chemokine ligand-16 (CXCL16) is a chemokine that exists either in its soluble form to bind to immune cells that express its receptor or in its native form to drive such immune cells to the site of inflammation [236]. Of note, ADAM10 and/or ADAM17 can cleave CXCL16 to its soluble form [212]. Considering that CXCL16 levels are elevated in T2DM patients compared to healthy individuals [237,238] and there is a marked upregulation of serum CXCL16 levels in β-cells in a T1DM model [232], we postulate that there will be increased ADAM10 and/or ADAM17 levels under such conditions. In support, Herman-Edelstein et al. [239] reported elevated ADAM10 and ADAM17 mRNA levels in diabetic hearts.

While the impact of diabetes on ADAM9, ADAM10, and ADAM17 expression remains under investigation, these family members have emerged as important contributors to SARS-CoV-2 infection. Their ability to cleave the viral S-protein (like TMPRSS2) can facilitate cellular entry. Despite the potential influence of diabetes on ADAM expression, their presence offers a level of redundancy in the viral entry process. This highlights the vulnerability of the virus, as targeting a single protease as a therapeutic intervention may be less effective. Future strategies should therefore focus on broader inhibition of multiple host cell proteases like ADAMs and TMPRSS2 to block SARS-CoV-2 entry into host cells more effectively.

### 4.7. Furin

As a transmembrane endoprotease, Furin cleaves various proproteins prior to secretion and as part of their maturation process [240]. As a proprotein convertase, Furin cleaves and activates many viral proteins, such as SARS-CoV-2 S-proteins, to facilitate its entry into host cells [241]. This cleavage occurs either during viral production in an infected cell or on virus entry into host cells [75,78,242]. The S-protein of SARS-CoV-2 contains a polybasic Furin cleavage site at the S1/S2 junction, which is absent in other SARS-CoVs [243]. The insertion of similar polybasic Furin cleavage sites increases virulence [244]. Based on the structure of this Furin cleavage site, it can be suggested that SARS-CoV-2 mimics host cell machinery for viral entry [245].

Here, an in vitro study showed that Furin KO decreased viral production by 100-fold [242], while others demonstrated that loss of the Furin cleavage site resulted in lowered viral pathogenicity [246]. Furin-mediated cleavage of this S1/S2 junction may only be valid in cells where the membrane fusion pathway prevails over the endocytosis entry pathway. This observation is based on results from Hoffmann et al. [75], where these authors observed how variants of SARS-CoV-2 S-proteins that were resistant to S-protein cleavage displayed significantly diminished entry into TMPRSS2-positive cells but not TMPRSS2-negative cells.

The expression of Furin in human pancreatic endocrine cells (including in β-cells) is moderate to high [30,147,247]. Furin controls proliferation and differentiation in islet cells [248] while playing a role in secretory granule acidification [249], with this being crucial for β-cell granule maturation and proinsulin-to-insulin conversion.

Increased Furin levels are associated with T2DM [250,251,252]. Moreover, elevated serum Furin levels can serve as a marker for diabetes progression and are correlated with metabolic dysregulations and an increased risk of diabetes-associated death [253]. This is mainly attributed to increased levels of osteopontin (pro-inflammatory glycoprotein) in diabetic individuals, and that can then upregulate Furin expression [253]. As a result of such a correlation between Furin and diabetes, it is plausible that this can aid SARS-CoV-2 entry by increasing viral entry and load, resulting in a poor prognosis for diabetic COVID-19 patients.

The intricate process of SARS-CoV-2 cell entry via the membrane fusion pathway is contingent on a complex interplay of viral and host proteins. The S-protein is pivotal for viral attachment and membrane fusion. Host cell receptors like ACE2 and other potential host factors (DPP4, NRP1, and GRP78) serve as the initial attachment points for the virus. Subsequent proteolytic processing of the viral S-protein by host proteases (TMPRSS2, ADAMs proteins, and Furin) is essential for membrane fusion. While these key proteins are currently being extensively characterized, the impact of comorbidities, particularly hyperglycemia, on their expression and function remains an area of active investigation. The potential influence of hyperglycemia on these proteins is summarized in Table 2. A comprehensive understanding of these molecular interactions is essential for developing effective therapeutic interventions.

## 5. Receptor-Mediated Endosomal Entry Pathway

Like other viruses, SARS-CoV-2 also uses host endosomal routes for viral entry [254]. Although the membrane fusion entry pathway is significantly more efficient than the endocytosis pathway, the viral entry mode amongst different cells depends on protease expression [255,256]. In the absence of TMPRSS2, SARS-CoV-2 is transported into the host endosome via endocytosis, either through clathrin-dependent or clathrin- and caveolae-independent entry pathways, for uptake and transport before they fuse with the lysosomes [257]. This is the case for SARS-CoV viral entry routes [257,258]. SARS-CoV-2, in particular, makes use of clathrin-mediated endocytosis to infect cells [259]. Endolysosomal fusion also activates lysosomal proteases (i.e., cathepsins), which cleave the S-protein to facilitate the fusion of the viral and host membranes to allow for the entry of viral RNA [260,261].

### 5.1. Cathepsins B/L

Cathepsins are non-specific proteases with endopeptidase and exopeptidase activities that participate in protein degradation in late endosomes and lysosomes. Cysteine proteases (Cathepsin B, L, and S) can contribute the most to viral entry by mediating the conversion of the virion into an infectious subvirion via partial proteolysis [262,263,264,265].

Cathepsin B is a lysosomal cysteine primarily involved in degrading or processing lysosomal proteins. In certain conditions, it is also involved in cell invasion, vesicle trafficking, inflammasome formation, and cell death [266]. However, Cathepsin L is integral in degrading extracellular, cytoplasmic, and nuclear proteins. It is also involved in many processes, including autophagy, apoptosis, cell cycle regulation, bone resorption, antigen processing, and tumor invasion/metastasis [267].

The role of cathepsins in viral entry is derived mostly from studies involving reoviruses, Ebola virus, and SARS-CoV [263,268,269] with limited studies on SARS-CoV-2 [270]. Cathepsin B plays an essential role in Ebola viral entry, whereas Cathepsin L plays a larger role in SARS-CoV [269,271,272] and SARS-CoV-2 [59,270] entry. This does not, however, exclude the possible role that Cathepsin B may play in terms of SARS-CoV-2 entry.

For example, Jaimes et al. [76] found that Cathepsin B can cleave SARS-CoV-2 S-proteins by using a biochemical peptide cleavage assay, while others found that Cathepsin B gene expression levels were upregulated in patients with severe COVID-19 [273]. Moreover, selective Cathepsin B inhibitors impaired SARS-CoV-2 infection in vitro, strengthening the notion that it plays an important role in SARS-CoV-2 entry [274]. However, conflicting data question the role of Cathepsin B in SARS-CoV-2 viral entry. Here, the authors showed that Cathepsin L was critical for S-protein activation by employing an S-protein pseudovirus system [275,276]. This indicates that SARS-CoV-2 may have a preference for Cathepsin L rather than Cathepsin B in terms of its entry mechanisms. However, further research work is required to better elucidate this vexing question.

Cathepsin L levels were increased following SARS-CoV-2 pseudovirus entry into a human hepatoma cell line (Huh7) [276]. The authors also found that Cathepsin L overexpression increased pseudoviral entry, while its knockdown and ACE2 inhibition in humanized ACE2 mice decreased SARS-CoV-2 entry. Moreover, Cathepsin L inactivation attenuated SARS-CoV-2 entry, although its inhibition was weaker in TMPRSS2-positive versus TMPRSS2-negative cells [59]. This indicates a compensatory relationship between TMPRSS2 and Cathepsin L, where the inactivity of one compensates for the other.

Cathepsin L cleaves the S-protein S1/S2 site (which differs from the TMPRSS2 cleavage site [241]) to enhance the entry of SARS-CoV-2. Cathepsin L is also secreted under systemic or local inflammatory conditions [277]; for example, Zhao et al. [276] reported increased circulating Cathepsin L levels in COVID-19 patients that positively correlated with disease severity. This indicates that circulating Cathepsin L may exacerbate viral entry during the inflammation that accompanies COVID-19.

Several studies revealed that TMPRSS2 (a fusion of the viral and host membrane) and Cathepsins B/L (a fusion of the viral and endosomal membrane) are dissimilar and work independently. Multiple studies reported inhibition of viral entry through the combined usage of TMPRSS2 and cathepsin inhibitors [59,256,274,278,279]. This emphasizes the importance of not only TMPRSS2 but also cathepsins in terms of SARS-CoV-2 entry into host cells.

Previous studies could detect Cathepsin B from isolated islets of Langerhans [280] and islet endocrine cells from the rat pancreas [281]. Others also identified Cathepsin B in insulin-secreting cell lines, such as HIT T15 [282] and INS-1 [283]. Furthermore, significant quantities of Cathepsin B mRNA were detected in total RNA from isolated islets of Langerhans but not in equivalent amounts of RNA from the whole pancreas [284]. In contrast, Cathepsin L is present in all pancreatic cell types [30,147] and is moderately expressed in pancreatic endocrine cells (i.e., α- and β-cells) [105].

Vidotti et al. [285] reported that high glucose levels stimulated the mRNA levels of Cathepsin B, with Liu et al. [286] reporting similar findings of increased Cathepsin B expression under high glucose stimulation. Contrastingly, Peres et al. [287] found that exposure to high glucose had no effect on its activity, while others found decreased Cathepsin B levels in the pancreatic β-cells of T2DM patients [288].

Cathepsin L expression in T2DM donors was also higher than in non-diabetic donors [105], while it was required to develop T1DM in mouse models [289] and regulate human and mouse islet cell proliferation [290]. These findings confirm a relationship between the presence of diabetes and greater Cathepsin L expression levels. There is also a relationship between Cathepsins B/L and hyperglycemia in the pancreas, with Jung et al. [283] reporting that Cathepsin B/L inhibition induced lysosomal dysfunction which enhanced pancreatic β-cell apoptosis under high glucose.

Research studies completed thus far have been unable to resolve conflicting findings regarding the effect of diabetes on Cathepsin B/L expression, and this requires further investigation. Regardless, Cathepsins B/L contribute to a complementary route for viral entry into endosomes. Such an alternative pathway highlights the adaptability of SARS-CoV-2 and the potential importance of this mechanism, particularly in instances where surface proteases are relatively limited. Thus, further research is needed to understand how diabetes may influence such cathepsins in terms of expression and function and also to ascertain their role in terms of SARS-CoV-2 infection.

### 5.2. Basigin

Basigin is a transmembrane glycoprotein that is a member of the immunoglobulin superfamily [291] and is also known as extracellular matrix metalloproteinase inducer (EMMPRIN) or CD147 [292]. It plays an important role in pathologies such as stroke, heart disease, and Alzheimer’s disease, and in the development, progression, metastasis, and prognosis of some human cancers [293,294]. Its role in infections by pathogens such as human immunodeficiency virus, hepatitis B and C viruses, and Kaposi’s sarcoma-associated herpesvirus is well studied, revealing associated mechanisms in terms of viral pathogenesis [295]. These findings allude to its possible role in SARS-CoV-2 pathogenicity.

Basigin was first highlighted as a SARS-CoV-2 S-protein co-receptor by Wang et al. [296], where the authors demonstrated that S-protein binding to Basigin elicited important functional implications in terms of viral entry via endocytosis. Since then, numerous studies revealed Basigin as a possible SARS-CoV-2 co-receptor [136,297,298,299,300,301,302,303,304,305,306,307,308]. Moreover, some also highlighted its potential role in the viral entry of SARS-CoV-2 into host cells with a relatively low abundance of ACE2 [309].

Basigin expression is also upregulated in various cell types upon ER and oxidative stress [292,310], both of which could occur in COVID-19 patients [178] and could propagate viral infection. Additionally, others detected Basigin in the pancreas of patients with COVID-19 [163]. Basigin is also a receptor for cyclophilin A (CyPA) [311], a member of the immunophilin family that is crucial in promoting viral infections. Here, CyPA is involved in viral invasion by human immunodeficiency virus-1 [312] and SARS-CoV [313], with their ability to infect host cells depending on CyPA and Basigin interactions. Such interactions occur when the viral N-protein binds to CyPA which in turn recognizes Basigin on the host cell membrane [313] to allow for the internalization of the virus.

Basigin is also expressed in the pancreas, with some studies reporting that it is highly expressed in α-, β-, and γ-cells [147,184,314], while others did not detect it in the islets [315]. CyPA was detected in pancreatic β-cells and its expression increased in high glucose-stimulated β-cells [316]. Moreover, these authors demonstrated how CyPA knockdown also enhanced insulin secretion and decreased β-cell apoptosis.

Basigin becomes active when glycosylated and participates in several physiological processes [317]. Due to its inherent glycosylation properties, Basigin is a candidate for modifications caused by hyperglycemia or by excess AGEs accompanying diabetes and metabolic diseases. In agreement, Basigin expression was upregulated by relatively high glucose levels and AGEs [302,318,319] and hence may play a crucial role in priming virus binding to host cells during diabetes. Mahmoud et al. [320] further confirmed this by reporting on the increased expression of Basigin in response to relatively high glucose levels in primary human preadipocytes. The increased expression of Basigin in response to high glucose levels may promote SARS-CoV-2 entry into the host cells, which may help explain the relatively high mortality rate in diabetic COVID-19 patients. The role of CyPA in SARS-CoV-2 infection may also be of importance in the diabetic context.

However, studies question the role of Basigin as a SARS-CoV-2 co-receptor. For example, Ragotte et al. [321] found that it did not directly interact with SARS-CoV-2 S-proteins. Shilts et al. [322] further supported this finding, showing no direct interaction between Basigin and the full-length S-protein or its S1 subunit. Thus, the role of Basigin in SARS-CoV-2 pathogenesis remains unclear and requires further investigation.

### 5.3. Aminopeptidase N

Aminopeptidase N (APN) (or CD13) is also referred to as a ‘moonlighting enzyme’ due to its multiple functions [323]. Here, several studies found that APNs execute crucial regulatory functions during normal and pathologic immune responses [324,325,326]. For example, it plays a role in antigen processing [327,328], cell trafficking [329,330,331,332], and the processing of inflammatory mediators, all of which are crucial features of immune responses. It is also involved in peptide cleavage, viral infection, endocytosis, and cell signal transduction [333].

APN has been identified as a viral receptor for other CoVs such as the porcine respiratory CoV, porcine transmissible gastroenteritis virus, feline infectious peritonitis virus, feline enteric CoV, and canine CoV [334,335]. These data indicate that APNs may also be involved in SARS-CoV-2 host entry [336,337]. Although some dispute this [57,59], we postulate that APN is indeed a receptor. In support of our argument, another CoV (HCoV-229E) uses APN as a receptor and utilizes the endosomal pathway for cellular entry [338].

It is also involved in an alternative RAAS axis that includes aminopeptidase A (APA) and insulin-regulated aminopeptidase (IRAP). Ang2 is cleaved by APA into Ang3, which is then cleaved into Ang4 by APN and bound to IRAP [339]. IRAP serves multiple functions in different organs (e.g., adipocytes and muscle cells) where it can co-localize with GLUTs. For example, in response to insulin, it can be redistributed from the endosome to the cell surface of GLUT4 specialized vesicles [340]. Of note, this translocation is impaired in T2DM patients [341]. Others also link the APN/Ang4/IRAP axis to glucose homeostasis, where there was decreased basal and insulin-stimulated glucose uptake into muscle cells and adipocytes in IRAP-deficient mice [342].

As APN is expressed in the pancreas [343,344] and IRAP is expressed in several cell types (liver, kidney, lung, pancreas, and neurons) [345], a postulate can be made that the APN/Ang4/IRAP axis plays a similar role in pancreatic glucose homeostasis. In agreement, some researchers detected GLUT4 expression in human pancreatic islets [346], while Härdtner et al. [115] found an activation of the APN/Ang4/IRAP axis in pancreatic β-cells under hyperglycemic conditions. In support, others also reported that Ang4 stimulates insulin signaling and secretion [347,348].

Thus, it is our opinion that a greater understanding of the potential interaction between APN and SARS-CoV-2, and particularly the role of the APN/Ang4/IRAP axis in the context of hyperglycemia, may lead to valuable insights regarding viral entry mechanisms and susceptibility. Further research is, therefore, crucial to determine whether APN plays a meaningful role in terms of SARS-CoV-2 infection and if its regulation under hyperglycemic conditions offers a potential target for therapeutic interventions.

### 5.4. Early Endosomal Antigen-1 and Rab Proteins

Early endosomal antigen-1 (EEA1) is a crucial component of the endosomal fusion process [349,350]. Rab proteins regulate endocytic pathways and belong to the Ras superfamily of small GTPases [351] that regulate intracellular material transport and innate immunity [352,353]. Of interest, Rab5 is considered the master regulator of endocytic trafficking and regulates the transport of newly endocytosed vesicles from the plasma membrane to early endosomes [354] and the membrane association of EEA1 [355,356,357]. It participates in the formation of clathrin-coated vesicles, the fusion of clathrin-coated vesicles with early endosomes, and the fusion between early endosomes [358,359]. Rab7 regulates early endosome maturation and trafficking from late endosomes to lysosomes [360]. Rab9 can promote late endosome entry into the trans-Golgi network [361], while Rab11 is involved in vesicle cycling [362].

Various viruses often use Rab proteins, and hence EEA1, to regulate the viral endosome transport process. These include the human immunodeficiency virus [361], herpes simplex virus [363,364], classical swine fever virus [365], CoVs (including porcine delta-CoV [366]), and porcine epidemic diarrhea virus [367]. In addition, beta-CoVs, including SARS-CoV-2, utilize lysosomes for egress [368].

Although the role of Rab proteins in CoV infection remains unclear, the knockdown of early endosome-related genes (EEA1 and Rab5), late endosome-related genes (Rab7A and Rab7B), and late endosome to lysosomal maturation-related genes significantly inhibited the entry of the mouse hepatitis virus into host cells [260]. Furthermore, Atik et al. [369] showed that Rab5 expression was significantly increased in COVID-19 patients, indicating that SARS-CoV-2 entry into host cells may be utilizing host early endosomes and that EEA1 may be a marker of this process. Others also showed that SARS-CoV-2 can enter host cells through clathrin-mediated endocytosis and that the genome is trafficked to early endosomes in a Rab5-dependent manner [370]. In agreement, COVID-19 patients displayed increased Rab5 expression and decreased Rab7 and Rab11B expression [369]. Others also found that the EEA1 gene residing in an associated locus linked to COVID-19 disease severity suggests an overactive cellular response that may facilitate SARS-CoV-2 entry and processing, thereby contributing to COVID-19 severity [371]. These findings indicate that SARS-CoV-2 can utilize the host cell’s early endosomes to enter. However, more research is required to fully elucidate the full picture in this instance.

EEA1 is expressed in the cytosol, endosome, and plasma membrane in various organs, including the pancreas [372]. For example, Jung et al. [373] detected Rab5, EEA1, and clathrin expression in a rat insulinoma cell line (INS-1E). Moreover, others found Rab7A in β-cells, Rab5 and Rab7 in human pancreatic cell lines, and Rab11 in a MIN6 insulin-secreting cell line [374,375,376].

Of note, Rab5/7/11 are all involved in insulin secretion; for example, Rab5 overexpression upregulated both glucose-stimulated insulin secretion and the GLP-1 receptor-mediated potentiation of insulin secretion in cultured β-cells [377]. GLP-1 can improve the glucose sensing, proinsulin biosynthesis, survival, and proliferation of β-cells [378,379].

Rab7 interacting lysosomal protein (RILP) acts as a downstream effector for Rab7 and is responsible for stabilizing Rab7 on the endosomal membrane [380,381]. RILP regulates insulin secretion via the mediation of lysosomal degradation of proinsulin in a Rab7-dependent manner [382]. These authors also demonstrated that RILP was upregulated in diabetic rat and mouse models, implying a potential pathogenic role in metabolic diseases such as DM. In agreement, some found that the inhibition of Rab7A may promote β-cell survival in the context of metabolic stress and prevent the onset of T2DM [375].

Rab11 participates in insulin granule exocytosis in β-cells [374] and is also involved in the endosomal recycling, sorting, and exocytotic GLUT4 movement in cardiac muscle [383]. It will be interesting to determine whether it plays a similar role in pancreatic β-cells.

Thus, EEA1 and Rab proteins play a crucial role in the intracellular trafficking and fusion of endosomes, a potential entry pathway for SARS-CoV-2. A greater understanding of how diabetes may influence the expression/activity of such proteins should provide valuable insights into alternative entry mechanisms, particularly in scenarios where the expression of surface proteases is limited. Further research is therefore necessary to determine whether hyperglycemia directly disrupts the regulation of EEA1 and Rab proteins, potentially affecting SARS-CoV-2 endosomal entry and susceptibility, especially in diabetic patients.

### 5.5. Valosin-Containing Protein

Valosin-containing protein (VCP) is a stress response protein involved in cell death and survival and is also referred to as ATPase p97 [384]. VCP can function as an apoptotic regulator and is an essential target in Akt signaling cascades [385]. It also functions in several other cellular functions, such as ER-associated degradation (ERAD), mitochondrial-associated protein degradation [386,387], the ubiquitin–proteasome system (UPS) [388], DNA replication and break repair [389,390,391], nuclear factor kappa-light-chain-enhancer of activated B cells (NF-κB) activation [392,393], and endomembrane fusion [394].

VCP also plays a role in endosomal trafficking [395,396], where it mediates endosome fusion through binding to clathrin, syntaxin, and EEA1 [394,397]. It can also control the EEA1 oligomeric state to regulate endosome fusion and trafficking [394]. This indicates a potential relationship between VCP and EEA1.

VCP also participates in multiple stages of virus production, including entry and uncoating, intracellular trafficking, nucleic acid replication, and egress [388,389,398,399,400,401,402,403]. Regarding its role in CoVs, VCP depletion decreased HCoV-229E replication in a Huh7 hepatoma cell line [404]. These authors also demonstrated how VCP silencing inhibited early-stage degradation of viral N-proteins and the consequent accumulation of virion-associated N-proteins. Since VCP is linked to the UPS, the accumulation of N-proteins could indicate that VCP is directly required for N-protein degradation. VCP inhibition also triggers antiviral effects, especially during the early stages of the viral life cycle, during virus uncoating and the RNA replication of HCoVs (HCoV-229E and HCoV-OC43) [405]. This suggests that VCP plays a role in maturing endosomes loaded with viral material for CoV replication. Research by Ramanathan et al. [394] and Meyer et al. [388] further confirms this.

Although it is evolutionarily conserved and ubiquitously expressed [397], VCP is predominantly localized in the cytosol while also found on ER membranes, the Golgi apparatus, the nucleus, mitochondria, and endosomes. Moreover, Nicolls et al. [406] found that it is expressed in normal adult mouse islets, while others detected VCP in the rat insulin-producing (RINm5F) cell line and human pancreatic islets [384]. Of note, the Human Protein Atlas [407] indicates that VCP protein levels in pancreatic endocrine cells are relatively low as it is mostly expressed in pancreatic α-cells.

Focusing on insulin synthesis, proinsulin molecules initially translocate into the ER lumen and then the Golgi apparatus, where they are subsequently processed into insulin and C-peptides [408]. Proinsulin may also enter the ERAD and undergo degradation in pancreatic β-cells [409]. As pancreatic β-cells from T1DM patients display relatively higher degrees of ER stress [410], ERAD proteins such as VCP are upregulated [411]. We therefore postulate that such accumulated substrates will be targeted for degradation and therefore not undergo processing to insulin. In support, VCP is associated with a model of T2DM [412] and is an ERAD protein that plays a crucial role in proinsulin degradation.

High glucose can trigger GLUT2 degradation in lysosomes in insulin-secreting Min6 cells [413]. GLUT2 mediates the entry and exit of glucose in organs such as the pancreas [414] and is integral for insulin secretion from β-cells. In agreement, Guillam et al. [415] found that GLUT2-KO mice exhibited diminished glucose clearance and plasma insulin levels due to impaired glucose-stimulated insulin secretion. GLUT2 is expressed on the apical membrane of β-cells [416,417], but lower levels are found in the diabetic pancreas [414]. Considering that EEA1 is an early endosome marker [418] and VCP facilitates protein degradation via lysosomal degradation [419], we propose that both proteins are involved in GLUT2 degradation and would be expected to be upregulated under high glucose conditions in pancreatic β-cells.

While VCP is known to be involved in various cellular processes, its specific contribution to viral entry or replication for SARS-CoV-2 needs further exploration. Here, an understanding of how hyperglycemia may influence VCP function could provide insights into potential mechanisms affecting viral pathogenesis in diabetic patients. Further research is, therefore, crucial to elucidate the specific role of VCP in SARS-CoV-2 infection and to determine if its regulation under hyperglycemic conditions presents a target for therapeutic interventions.

SARS-CoV-2 can also enter host cells via endocytosis. This pathway necessitates a complex interplay between viral and host factors. Upon internalization, the virus is trafficked through the endosomal compartment, where acidic conditions facilitate viral entry. Endosomal proteases such as Cathepsins B/L are crucial for viral S-protein processing. Additionally, host proteins (Basigin, APN, EEA1 together with Rab5/7/11, and VCP) possibly play pivotal roles in endosomal trafficking and viral uncoating. The precise mechanisms underlying the impact of hyperglycemia on these proteins and subsequent viral entry remain to be fully elucidated, but emerging evidence suggests potential alterations in protein expression and possibly activity. A comprehensive understanding of these factors is essential for developing targeted therapeutic interventions. The following Table 3 summarizes the key proteins involved in the endosomal entry of SARS-CoV-2 and explores the potential impact of hyperglycemia on them.

## 6. Conclusions

As the specific mechanisms of SARS-CoV-2 host cell entry are becoming increasingly understood, it remains unclear how pre-existing conditions like DM and hyperglycemia can worsen COVID-19 outcomes. This review highlighted the potential interplay between hyperglycemia, SARS-CoV-2 entry, and accessory proteins in pancreatic β-cells. Current research reveals how hyperglycemia is potentiating the viral entry of SARS-CoV-2 into pancreatic β-cells, resulting in their dysfunction and inability to produce insulin. Further research is warranted to elucidate the specific mechanisms involved and explore potential therapeutic targets to mitigate the detrimental effects of hyperglycemia on COVID-19 pathogenesis in diabetic individuals.

## Figures and Tables

**Figure 1 viruses-16-01243-f001:**
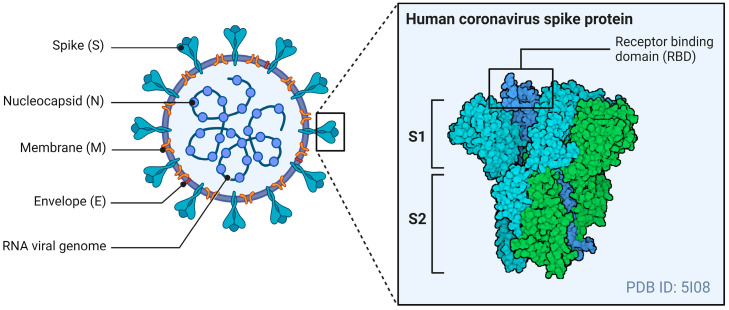
The general structure of a CoV. The S-protein of an HCoV, containing the RBD, mediates membrane fusion by binding to various cellular receptors. Adapted from “*Coronavirus Structure and Protein Visualization*”, by Biorender.com (2024). Retrieved from https://app.biorender.com/biorender-templates (accessed on 24 May 2024).

**Figure 2 viruses-16-01243-f002:**
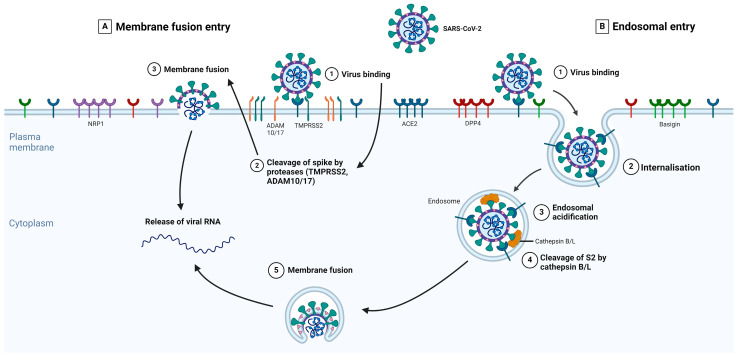
The expanded entry pathways of SARS-CoV-2. The S-protein of SARS-CoV-2 binds to its primary receptor, ACE2, or potentially to Basigin (step 1), which induces conformational changes in the S1 subunit and exposes the S2 site for cleavage. The S-protein can bind to NRP1 and DPP4, enabling membrane fusion. Depending on the entry pathway taken by SARS-CoV-2, the S2 site is cleaved by either TMPRSS2 or ADAM 10/17 (membrane fusion entry, part A) or Cathepsins B/L (receptor-mediated endosomal entry, part B). In the presence of proteases on the plasma membrane, S-protein cleavage occurs at the cell surface (part A, step 2). With endosomal entry, the S-protein-receptor complex is internalized via clathrin-mediated endocytosis (part B, step 2) into endolysosomes, where the S2 site is cleaved by Cathepsins B/L (part B, step 4). Before this, the cleavage via cathepsins requires an acidic environment for its activity; therefore, endosomal acidification occurs (part B, step 3). Fusion between the virus and host cellular membranes forms a fusion pore where viral RNA is released into the host cytoplasm for viral uncoating and replication. Adapted from Jackson et al. [79]. Created in BioRender.com.

**Table 1 viruses-16-01243-t001:** Studies reporting on NOD cases.

Main Findings	Country	Study Design	Reference
A total of 29.1% (605 *) of patients presented with NOD.	China	Retrospective	[11]
A total of 21% (453 *) of patients were newly diagnosed with diabetes (fasting admission glucose ≥ 7.0 mmol/L and/or HbA_1c_ ≥ 6.5%).	China	Retrospective	[14]
A total of 34.3% (233 *) of patients had diabetes. Among them, 45% (33 ^1^) were newly defined as having undiagnosed diabetes (HbA_1c_ level ≥ 6.5% or 48 mmol/mol) at admission.	China	Retrospective	[19]
A total of 91% (33 *) of children presented with new-onset T1DM; 5 children tested positive for SARS-CoV-2.	U.K.	Cross-sectional	[20]
A total of 9.8% (64 *) of patients presented with new-onset T1DM, and 15.6% (33 ^1^) in the COVID-19-positive group also presented with NOD.	U.S.	Cross-sectional	[21]
A total of 13.3% (3711 *) of COVID-19 patients had newly diagnosed diabetes; a random effects meta-analysis estimated a pooled prevalence of 14.4% (95% CI: 5.9–25.8%) for NOD.	U.S., Italy, China	Systematic review and meta-analysis	[22]
A total of 5.1% (413 *) of patients presented with newly detected diabetes.	Italy	Retrospective	[23]
A total of 5.7% (35 *) of COVID-19 patients were new presentations of diabetes.	U.K.	Retrospective case series	[24]

Abbreviations: T1DM—type 1 diabetes mellitus; CI—confidence interval. * Total number of participants in the study. ^1^ Number of participants in a group within the study.

**Table 2 viruses-16-01243-t002:** A summary of the potential influence of hyperglycemia on the proteins that mediate the membrane fusion entry of SARS-CoV-2.

Membrane Fusion Entry Protein	Effect of Hyperglycemia	References
ACE2	Increased ACE2 glycosylation resulting in loss of interaction between ACE2 and SARS-CoV-2 S-protein	[118,119]
DPP4	High levels in DM patients; increased mRNA expression and activity in hyperglycemia	[156,157,158]
NRP1	Increased RNA expression in diabetic kidney; reduced NRP1 expression in diabetic nephropathy patients	[167,168,169]
GRP78	Increased GRP78 levels in diabetic mice and T2DM islet donors; increased serum concentration of GRP78 in T2DM patients; no effect; increased ER stress-mediated GRP78 expression in hyperglycemia	[185,186,187,188,189,190]
TMPRSS2	High expression in diabetic COVID-19 heart autopsies; no effect	[118,204]
ADAM9/10/17	Increased ADAM9 expression; increased ADAM10 and ADAM17 mRNA expression; increased CXCL16 (ADAM10/17 substrate) cleavage in diabetes.	[232,233,234,235,237,238,239]
Furin	Increased Furin levels in T2DM	[247,251,252,253]

**Table 3 viruses-16-01243-t003:** A summary of the potential influence of hyperglycemia on the proteins that mediate the endosomal entry of SARS-CoV-2.

Endosomal Entry Protein	Effect of Hyperglycemia	References
Cathepsins B/L	Increased mRNA expression of Cathepsin B; no effect; decreased Cathepsin B in β-cells of T2DM patients; increased Cathepsin L expression in T2DM donors	[105,285,286,287,288]
Basigin	Expression upregulated by high glucose and AGEs	[302,318,319,320]
APN	Activation of APN/Ang4/IRAP axis under hyperglycemia	[115]
EEA1 and Rab proteins	RILP (associated with Rab7) upregulated in diabetic rat and mouse models	[382]
VCP	Upregulated levels in T1DM patients due to ER stress	[411]

## Data Availability

Data sharing is not applicable.

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
