# Peer review of "Potential Effects of Hyperglycemia on SARS-CoV-2 Entry Mechanisms in Pancreatic Beta Cells"

_viruses, 2024, doi:10.3390/v16081243_

Round 1

Reviewer 1 Report

Comments and Suggestions for Authors

SARS-CoV-2 emerged in late 2019 and was responsible for the devastating COVID-19 pandemic.  The virus causes a wide range of disease severities, ranging from relatively mild illness to severe disease that can be fatal; the pandemic was responsible for 7 million deaths worldwide.  Indeed, despite a highly effective vaccination program, the virus is still a concern due, in part, to the continued emergence of novel variants.  Among patients that experience the most severe symptoms, those with underlying comorbidities predominate.  These conditions include diabetes, cardiopulmonary disease and obesity, among others.  There is no doubt whatsoever that these conditions impart a significantly greater risk for hospitalization and death from COVID-19.  This review addresses the mechanism by which SARS-CoV-2-induced disease is exacerbated by one these major comorbidities, diabetes.  In particular, it explores the relationship between the hyperglycemia that characterizes diabetes mellitus and the ability of the virus to gain entry into the pancreatic b-cells, the dysregulation of which is the hallmark of diabetes.

The entry of SARS-CoV-2 into cells is remarkably complex, depending in large part on the particular cell type and the available potential receptors therein.  The virus can enter cells by either a membrane fusion or receptor-mediated endosomal entry pathway.  This review presents a detailed treatise on the relationship between hyperglycemia and SARS-CoV-2 entry, focusing on the interplay between the virus and the array of receptors and accessory proteins associated with each of the avenues of entry.  Indeed, this relationship is clearly a two-way street, not only does diabetes exacerbate SARS-CoV-2 disease, but the virus can also accelerate the progression and severity of diabetogenic effects.

A particularly alarming observation is the emergence of cases of new onset diabetes (NOD) in SARS-CoV-2 patients, which speaks to the potential for the direct infection of pancreatic b-cells by the virus and the inevitable exacerbation of hyperglycemia.  This possibility is further supported by reports of acute pancreatic disease in SARS-CoV-2 patients.  The bidirectional mutual enhancement of SARS-CoV-2 infection and diabetes easily explains this emergence of these NOD cases. 

This review focuses on the relationship between hyperglycemia and SARS-CoV-2 entry.  The authors truly get “into the weeds” of how hyperglycemia affects the multiple receptors and accessory proteins involved in both membrane fusion and receptor-mediated endosomal entry.  The authors take the reader one-by-one through each of the numerous SARS-CoV-2 receptors and associated accessory proteins, in the process painstakingly elaborating how hyperglycemia modulates the levels and functions of each and especially their interaction with the viral spike protein.  Whether it be ACE2, DPP4, neuropilin-1, GRP78, furin, cathepsins, TMPRSS2, Basigin or multiple other receptor/accessory proteins, the authors make an exceptionally strong case that the interplay between hyperglycemia and each these proteins in pancreatic b-cells modulates SARS-CoV-2 entry.  This is considered an exceptionally detailed and exquisitely comprehensive review of the topic.  It is recognized as an important contribution to our understanding of the relationship between diabetes and SARS-CoV-2 infection that moves the field forward in significant ways.  There are no detected weaknesses in either the presentation or interpretation of the data.  One noticeable terminology that follows the discussion of just about every receptor/accessory protein is some variation of the statement that our understanding of the relationship between hyperglycemia and the role of that particular factor in virus entry “requires further investigation”.  This review lays the groundwork for those studies.

Author Response

We thank the reviewer for the thorough review and kind and valuable feedback. 

Reviewer 2 Report

Comments and Suggestions for Authors

This manuscript described nicely, regarding COVID-19 and diabetes, indicating risk of complications after COVID-19 in diabetic patients but also the risk of diabetes development following SARS-CoV-2 infection. In addition, authors focused on the mechanisms of SARS-CoV-2 infection, mainly and significantly virus entry through virus receptor ACE to pancreatic beta-cells and also followed by the intracellular proliferation mechanisms through many candidate molecules. I admire your efforts to write this review article.

Author Response

We thank the reviewer for the time taken to review the manuscript and providing valuable feedback.

Reviewer 3 Report

Comments and Suggestions for Authors

1. Please double check the key references. For example, (1) line 117: SARS-CoV-2 enters host cells via its primary receptor, ACE2 [57]. Two other references ( PMID: 32142651 and PMID: 32015507) should be cited here; (2) from line 380 to 385, two key references (PMID: 33965375 and PMID: 37967509) should be included;…

2. Simplify the parts of Dipeptidyl peptidase-4, Aminopeptidase N, Basigin , Early endosomal antigen-1 and Rab proteins, and Valosin-containing protein.

3. Suggest to add some illustrations describing how hyperglycemia can influence proteins involved in membrane fusion, potentially enhancing the susceptibility of β-cells to infection.

Author Response

  1. Please double check the key references. For example, (1) line 117: SARS-CoV-2 enters host cells via its primary receptor, ACE2 [57]. Two other references ( PMID: 32142651 and PMID: 32015507) should be cited here; (2) from line 380 to 385, two key references (PMID:33965375 andPMID: 37967509) should be included;…

Response: We thank the reviewer for the time taken and kind feedback to improve the manuscript. We note the suggested additions to the references and amended/included references as follows: 

Table 1: Changed order of studies depicted in the table for refs to appear numerically

Line 104: Changed ref #43

Line 107: Refs added (56, 57, 58)

Line 150: Ref changed (68)

Line 179: Original ref 77 (Zhou et al., 2020) removed

Line 213: Refs added (57, 59)

Line 215: Ref changed (86)

Line 217: Ref changed (87)

Line 236: Original ref 32 (Muller et al., 2021) removed

Line 812: Original ref 177 (Rosa-Fernandez et al., 2021) removed

  1. Simplify the parts of Dipeptidyl peptidase-4, Aminopeptidase N, Basigin , Early endosomal antigen-1 and Rab proteins, and Valosin-containing protein.

Response: We thank the reviewer for the suggestion and evaluated the information in each of the mentioned sections. We are of the opinion that the information in these sections is important to comprehensively discuss the role of each, and hence have not edited the text. We have included two more tables in line with this comment and comment 3 below, to summarize the hyperglycemic effects on each of the marker discussed in the paper. We believe this will guide the reader to focus on some of the main proposed mechanisms in this regard. 

  1. Suggest to add some illustrations describing how hyperglycemia can influence proteins involved in membrane fusion, potentially enhancing the susceptibility of β-cells to infection.

Response: We thank the reviewer for this suggestion and agree that it would enhance the manuscript. After considering several potential options for illustrations, we decided that the complexity of the mechanisms may be better suited to summarizing as tables. We therefore included the text in lines 547-557 accompanying Table 2 (page 12) and the text in lines 844-855 accompanying Table 3 (page 18) to summarize the effects of hyperglycemia/diabetes on the membrane fusion and endosomal entry proteins, respectively.  

We further included additional edits:

Line 384-388: New information added in highlighted text.

Line 702: Changed text from “These data indicate” to “This data indicates”

All edits and additions are indicated in yellow highlight in the revised manuscript.